# Factors Associated with Vitamin D Testing: A Population-Based Cohort Study in Queensland, Australia

**DOI:** 10.3390/nu17152549

**Published:** 2025-08-04

**Authors:** Vu Tran, Donald S. A. McLeod, Catherine M. Olsen, Nirmala Pandeya, Mary Waterhouse, David C. Whiteman, Rachel E. Neale

**Affiliations:** 1QIMR Berghofer, Brisbane, QLD 4006, Australia; vu.tran@qimrb.edu.au (V.T.); donald.mcleod@qimrb.edu.au (D.S.A.M.); catherine.olsen@qimrb.edu.au (C.M.O.); mary.waterhouse@qimrb.edu.au (M.W.); david.whiteman@qimrb.edu.au (D.C.W.); 2School of Public Health, The University of Queensland, Brisbane, QLD 4006, Australia; 3Royal Brisbane and Women’s Hospital, Brisbane, QLD 4029, Australia; 4Medical School, The University of Queensland, Brisbane, QLD 4006, Australia

**Keywords:** vitamin D testing, population-based cohort, inappropriate testing, Australia, clinical indicators, vitamin D deficiency, healthcare utilization, 25-hydroxyvitamin D, general practitioner

## Abstract

**Background/Objectives**: Vitamin D testing has increased significantly in developed countries in recent decades. We aimed to describe trends in vitamin D testing rates and factors associated with testing and vitamin D deficiency in Queensland, Australia (2011–2019). **Methods**: We used data from the QSkin Sun and Health Study (*n* = 40,417), a prospective population-based cohort study with linkage to the Medicare Benefits Schedule, Pharmaceutical Benefits Scheme, and pathology laboratories. Main outcomes included age-standardized incidence rate of vitamin D testing; having ≥1 vitamin D test during follow-up; vitamin D deficiency (25-hydroxyvitamin D concentration <50 nmol/L) in the first vitamin D test; and repeat vitamin D tests. **Results**: The age-standardized incidence rate of testing increased by 2% per quarter during follow-up. Of the 35,250 participants analyzed for associations with testing (median age of 57 years, 52% female), 45% had ≥1 vitamin D test. Among those tested, 56% had no apparent clinical indication for their initial vitamin D test, 21% were vitamin D deficient in their initial test, and 58% had a repeat test. Repeat testing occurred in 56% who were not deficient in their prior test, while only two-thirds of those deficient received a follow-up assessment. Participants who visited a general practitioner ≥2 times in the year prior to follow-up were 60% more likely to have ≥1 vitamin D test compared with those with no visit, but general practitioner (GP) visits were not associated with risk of vitamin D deficiency. **Conclusions**: These results suggest that initiatives are needed to help clinicians target vitamin D testing in alignment with clinical guidelines.

## 1. Introduction

There has been a dramatic increase in the rate of vitamin D testing over the past two decades in Australia. Between 2000 and 2011, the rate increased nearly 100-fold [1], leading to concerns about over-testing. Similar trends have emerged across other developed countries, such as the United States [2], Canada [3], France [4], and the United Kingdom [5].

After a review of the evidence [6], in November 2014 the Australian Department of Health introduced new eligibility criteria for government-reimbursed vitamin D tests [7]. These criteria required that vitamin D testing be restricted to people at high risk of deficiency or with clinical indications, after which there was an immediate decrease in testing rates [8]. However, this trend was not sustained [9,10], and by late 2016 the rate had reverted to levels observed before the criteria were introduced [10].

Adherence to the criteria for testing may be suboptimal. A review of a large nationwide dataset (~1.5 million Australian adults) estimated that approximately three-quarters of vitamin D tests ordered by a general practitioner (GP) did not have an obvious clinical indication [8]. Moreover, the prevalence of test results indicating deficiency actually decreased after the introduction of the criteria [10]. Similar patterns of vitamin D testing have been observed in other developed countries [4,11], making Australia’s policy intervention experience relevant internationally.

Identifying factors associated with vitamin D testing may inform further initiatives to optimize the use of testing. An increased likelihood of vitamin D testing has been found among older people, women, migrants, those living in higher socioeconomic areas, and those who visit doctors more frequently [8,12,13,14]. However, no studies to date have investigated testing among individuals who might be at risk of vitamin D deficiency due to sun avoidance (e.g., people with a history of skin cancer), and few studies have explored the issue of repeat testing.

In this paper, we analyze trends in vitamin D testing and the percentage of tests showing vitamin D deficiency between 2011 and 2019 among a population-based cohort in Queensland, Australia, and identify factors associated with vitamin D testing and vitamin D deficiency.

## 2. Materials and Methods

### 2.1. Study Population and Data Collection

We used data from the QSkin Sun and Health Study (QSkin), a prospective cohort study established for the study of skin cancer [15]. Between 2010 and 2011, 43,794 residents of Queensland aged 40 to 69 years were recruited into the QSkin Study, using the Australian Electoral Roll as the sampling frame. The QSkin Study was approved by the QIMR Berghofer Medical Research Institute Human Research Ethics Committee (approval number P1309, approval date 21 May 2010).

At baseline all participants provided written consent, including consent to collect information from the Queensland Cancer Register (QCR), pathology laboratories, and hospitals. Participants were also asked to provide optional consent to linkage of their records with the Medicare Benefits Schedule (MBS) and Pharmaceutical Benefits Scheme (PBS). The MBS database holds information on all medical services provided in Australia outside the public hospital system, including in primary care. The PBS records information for prescription medications dispensed that receive a government subsidy (i.e., most commonly prescribed medications). Participants completed a baseline survey that asked about their sociodemographic, phenotypic, lifestyle factors and medical history, including the treatment of skin cancers.

### 2.2. Outcomes

The main outcomes were: (1) incidence of at least one vitamin D test during follow-up; (2) prevalence of vitamin D deficiency (serum 25-hydroxyvitamin D (25(OH)D) concentration <50 nmol/L) in the first vitamin D test; and (3) incidence of repeat vitamin D tests among participants with at least one vitamin D test. We also described temporal trends (2011 to 2019) in the rates of vitamin D testing and the percentage of tests indicating vitamin D deficiency.

To ascertain outcomes, we first identified all QSkin participants who received at least one government-funded vitamin D test, as recorded in the MBS dataset. For those participants, we then conducted data linkage with the two major private pathology laboratories in Queensland and obtained the results of all vitamin D tests. We matched MBS and pathology records by date and considered records within 30 days as part of a single testing episode. Testing episodes that could not be matched (i.e., those that had only MBS or pathology records), were included in the analyses.

### 2.3. Explanatory Variables

Details about the derivation of explanatory variables are provided in Appendix A. These variables included sociodemographic, phenotypic and lifestyle factors, self-reported history of skin cancers, melanoma diagnoses, prescription of medications that either interfere with vitamin D metabolism or are used to treat conditions associated with vitamin D deficiency, frequency of visits to GPs, unweighted Rx-Risk comorbidity index (a medication-based comorbidity index that quantifies chronic disease burden using prescription drug dispensing data) [16], and the result of any previous vitamin D test. We used Socio-Economic Indexes for Areas (SEIFA) to estimate socioeconomic status [17]. A skin phenotype score, as an indicator of skin cancer risk, was calculated based on phenotypic characteristics and categorized into tertiles. We coded medication prescriptions: as (1) ever prescribed; and (2) recently prescribed (within the last 90 days). Explanatory variables were ascertained from the baseline survey, MBS/PBS/pathology data, and the QCR.

### 2.4. Assigning an Indication to the First Vitamin D Test

We used the MBS testing criteria for which data were available (i.e., osteoporosis and antiepileptic medication, long-term glucocorticoid use, low sun exposure), and factors known to be associated with vitamin D deficiency (i.e., obesity) to assign the most likely indication for a participant’s first vitamin D test (indications that we were unable to ascertain are shown in Appendix A). The hierarchy in which we assigned the indication was as follows: recent prescription of osteoporosis/antiepileptic medication (i.e., first prescribed within 90 days before the test); long-term glucocorticoid use (dispensed at least 4 times in the 12 months before the test); low sun exposure at baseline; and being obese at baseline. We classified participants who did not satisfy any of the criteria as having no apparent indication for vitamin D testing. In a sensitivity analysis, we included Rx-Risk score ≥2 (at the time of the test) as another indication, inserting it between long-term glucocorticoid use and low sun exposure in the original hierarchy.

### 2.5. Follow-Up

Follow-up began one year after the date of consent. We used the 12-month period after consent to identify people who had ‘prevalent’ vitamin D tests and to derive explanatory variables based on MBS and PBS data. In analyses of associations with having ≥1 vitamin D test, follow-up ended at the earliest of: (1) first vitamin D test; (2) date of death; or (3) 31 December 2019. For analyses of repeat testing, follow-up ended at the earliest of date of death or 31 December 2019, except for analyses of the association with the result of the preceding test, where the date of the first vitamin D test for which the result was unknown was also used to censor participants.

### 2.6. Eligibility

We excluded participants who did not consent to linkage with the MBS and PBS, had an unknown date of death, or had dates of death preceding dates of vitamin D tests, suggesting incorrect data linkage. For analyses of factors associated with vitamin D testing and deficiency, we also excluded participants who had undergone a vitamin D test or died in the 12 months between consent and the start of follow-up. We decided a priori that sex, age, skin phenotype, and SEIFA were important potential confounders, so participants with missing data for any of these variables were excluded (*n* = 846, 2.1%). Participants for whom the 25(OH)D concentration was missing for the first test in the MBS data were excluded from the analysis of factors associated with vitamin D deficiency.

### 2.7. Statistical Analyses

We used the Joinpoint Regression Program (v5.0) [18] to analyze trends in vitamin D testing. Additionally, we described person-based rates of testing and the proportion of tests consistent with vitamin D deficiency by season during the follow-up period. We compared test-based incidence rates within those aged 45–74 years in the QSkin Study cohort with those in the Queensland population (using aggregated MBS data available online from Services Australia) [19].

All other analyses were performed in R v4.3.1. We used Cox regression, log-binomial regression, and Andersen–Gill models with robust standard errors to estimate associations with having a vitamin D test, vitamin D deficiency, and repeat testing, respectively. Analyses of factors associated with having a test were repeated according to time period (2011–2014 versus 2015–2019) (Appendix A). For analyses of repeat testing, all exposure variables, except sex, were time-varying, and all models included the number of preceding vitamin D tests. Directed acyclic graphs were used to guide our choice of adjustment variables.

## 3. Results

We included 40,417 QSkin participants in the analysis of trends in vitamin D testing, and 35,250 in analyses of factors associated with testing (Figure 1). The median age at cohort entry was 57 years, 52% were female, 94% were of White European descent, 15% reported low sun exposure (≤3.5 h/week), and 40% reported prior skin cancer excisions (Table 1). Test results were available from the pathology laboratories for 83% of testing episodes recorded in the MBS. We could not capture the results of vitamin D tests from all laboratories, resulting in 17% of MBS testing episodes lacking a test result.

### 3.1. Trends in Vitamin D Testing and Deficiency

Between 2011 and 2019, the person-based incidence of vitamin D testing was 602 per 10,000 person-years. On average, the age-standardized incidence rate of testing increased by 2.0% (95% confidence interval [CI] 1.6 to 2.6) per quarter between 2011 and 2019. Joinpoint regression identified three inflection points (Q1 2013; Q4 2014; Q3 2015) delineating four separate trend intervals, with average quarterly percent changes of 8.4% (95% CI 6.0 to 14.3), −0.5% (95% CI −3.9 to 3.5), −13.7% (95% CI −16.5 to 1.9), and 3.0% (95% CI 2.3 to 3.8), respectively (Figure 2, Appendix A). The test-based incidence rate in the QSkin cohort was similar to that in the Queensland population, and the high rate of testing has persisted in the Queensland population up to the end of 2023 (Appendix A). Sex-specific (Figure 2, Appendix A) and age-specific (Appendix A) trends were similar to the overall trend. Overall, 32% of participants had two or more vitamin D tests (Appendix A).

The median 25(OH)D concentration remained stable over time at approximately 70 nmol/L (Figure 3a). In 2011, 17% of the tests indicated deficiency, compared with 13% in 2019 (Figure 3b).

### 3.2. Factors Associated with Having a Vitamin D Test

Of the participants included in analyses of factors associated with testing, 45% (*n* = 15,745) had at least one vitamin D test during follow-up. We found no apparent indication for the first vitamin D test in 56% of those tested; obesity and low sun exposure were the most common indications (22% and 19% of those tested, respectively) (Table 2). When we considered Rx-Risk score ≥2 to be an indication in the sensitivity analysis, it became the most common indication (49% of those tested), and the percentage of tests with no apparent indication substantially decreased to 32% (Appendix A).

Rates of testing were generally higher in spring (Appendix A). The factors most strongly associated with testing were recent prescription of medication to treat osteoporosis (hazard ratio (HR), 5.3; 95% CI 4.4 to 6.5), and being female (HR, 2.5; 95% CI 2.4 to 2.6). Other factors associated with increased likelihood of testing were: increasing age; non-European ancestry; greater socioeconomic advantage; prescription of antiepileptic medication and menopausal hormone therapy (MHT); higher Rx-Risk index; and more GP visits. People with a more sun-sensitive phenotype were more likely to be tested and, while somewhat inconclusive, there was evidence of more testing in people diagnosed with skin cancer. People who spent more time outdoors were less likely to be tested, but there was no association with sunscreen use (Table 1). The strength of the associations with sex, age, ancestry, and socioeconomic status was somewhat attenuated after the introduction of MBS criteria in November 2014, whereas the associations with recent prescription of osteoporosis and antiepileptic medication were stronger (Appendix A).

### 3.3. Factors Associated with Being Vitamin D Deficient at the First Vitamin D Test

Among those for whom we obtained the result of the first vitamin D test (*n* = 12,952), the median 25(OH)D concentration was 65 nmol/L; 21% of participants were vitamin D deficient. The percentage deficient was higher among those with versus without an apparent indication for a test (28% vs. 16%) (Table 2). By season, the lowest 25(OH)D concentration and highest percentage of tests indicating deficiency were observed in winter, followed by spring (Appendix A). The factor most strongly associated with vitamin D deficiency was obesity (risk ratio (RR) 2.0; 95% CI 1.8 to 2.1). Other factors associated with increased risk of deficiency were being female, non-European ancestry, greater socioeconomic advantage, smoking, and prescription of antiepileptic medication (Table 3). There was a small increase in the risk of deficiency among those with a skin phenotype that increased their predisposition to skin cancers, but a decreased risk in those with a skin cancer diagnosis. Factors associated with a lower risk of deficiency were increasing sun exposure, sunscreen use, alcohol consumption, and use of osteoporosis medication (Table 3).

### 3.4. Factors Associated with Having a Repeat Vitamin D Test

Among participants who were tested, 58% (*n* = 9099) had at least one repeat test. The median time between consecutive tests was around 12 months overall (Appendix A) for most participant subgroups, but it was approximately 6 months following tests indicating moderate to severe vitamin D deficiency (less than 30 nmol/L) (Appendix A). Just over half (56%) of those not vitamin D deficient on the first test had a repeat test (median gap 12 months). Among those who were deficient, 69% had a repeat test, but it was after 12 months in 38% of these individuals.

The likelihood of repeat testing increased with vitamin D deficiency, female sex, advancing age, more GP visits, and a higher Rx-Risk index (Table 4).

## 4. Discussion

Vitamin D is crucial for musculoskeletal health [20], and has potential effects on cancer mortality [21,22], infection [23,24,25], and autoimmune disease [26,27]. It is thus important to avoid vitamin D deficiency. Vitamin D testing is advised for populations at high risk of deficiency as a case-finding strategy [28,29], but routine population screening is not recommended [30]. In this cohort of over 40,000 Australians, we found a relatively high incidence of testing, with nearly half (45%) of all participants having had at least one test between 2011 and 2019.

We found that rates of vitamin D testing followed patterns previously reported [9,14]. The late 2012 change in direction appears to reflect growing clinical and policy awareness about high rates of testing that preceded the formal policy intervention. Published evidence of overtesting emerged in 2012–2013 [1], and data show that testing rates were already declining before the formal November 2014 MBS criteria were introduced [9,14]. The introduction of MBS eligibility criteria in November 2014 initially reduced testing rates, but the subsequent bounce-back by mid-2015 demonstrates that these criteria alone were insufficient to sustain the reduction in vitamin D testing.

As shown previously [8,10], the prevalence of deficiency in tested samples decreased over time. The reason for this trend is unknown, but is most likely related to increased awareness of vitamin D deficiency and growing use of vitamin D supplementation in recent years. If true, this raises the question of whether initial vitamin D tests are needed in patients who are already taking vitamin D supplements.

In contravention of Australian government guidelines, 32% of participants had no apparent indication for their first test even when we considered factors strongly associated with vitamin deficiency but not included in the MBS criteria as indications for testing (i.e., obesity, co-morbidities). Additionally, repeat testing occurred in over half of those who were not deficient. These findings, along with the positive association between more GP visits and testing (without a corresponding increase in risk of deficiency), suggest that a degree of untargeted screening is occurring.

Australian guidelines recommend that individuals found to be deficient and initiated on supplementation should be retested within 3–4 months [29]. Only two-thirds of those deficient were retested, and approximately 38% occurred more than 12 months after the test identifying vitamin D deficiency, providing further evidence that testing is not always targeted appropriately.

The Royal College of Pathologists of Australasia and Royal Australian College of General Practitioners recommend vitamin D testing in people chronically lacking in sun exposure [28,29]. We found low sun exposure to be the apparent indicator for testing in approximately 19% of those tested. However, only 5–10 min/day outdoors is needed to maintain adequate vitamin D status in Queensland (provided sufficient skin is exposed), so many of these participants may not have been underexposed. A newly released position statement might be able to assist clinicians in identifying patients with low sun exposure [31] who would then be eligible for testing. Alternatively, supplementation could be initiated without testing according to Australian nutrition guidelines which recommend that those patients should consume 400–600 IU of vitamin D per day [32]. These guidelines are consistent with the United States Endocrine Society recommendations, which implicitly assume that people have sufficient sun exposure or supplementary vitamin D to avoid vitamin D deficiency [30]. Ensuring that individuals meet these guidelines may obviate the need for testing in many people. Additional research is needed to understand clinicians’ use of the low sun exposure criterion to triage patients for vitamin D testing and their perspective on supplementation versus testing.

We included Rx-Risk to reflect real-world clinical decision-making, recognizing that GPs may order vitamin D testing to investigate whether vitamin D deficiency could be contributing to patients’ chronic conditions or overall health status, or conversely, whether chronic health conditions might lead to reduced vitamin D. This approach accommodates clinical judgment beyond the specific MBS criteria, acknowledging that clinicians may consider vitamin D testing as part of a broader diagnostic workup for patients with multiple comorbidities. We positioned Rx-Risk (≥2) between long-term glucocorticoid use and lifestyle factors (low sun exposure, obesity) because it represents intermediate clinical risk. Specifically, we considered indications included in the MBS criteria (osteoporosis/antiepileptic drugs, glucocorticoids) as having the strongest clinical rationale, followed by chronic disease burden (Rx-Risk), and finally lifestyle/demographic factors, which were collected at baseline. This hierarchy reflects decreasing clinical specificity for vitamin D testing indications.

A limitation of this study is that data about vitamin D testing were only available in the QSkin cohort up to 2019. However, there has been no decrease in the rate of testing in Queensland in the past 5 years, suggesting that there remains a need to optimize testing to target those most at risk of vitamin D deficiency. In addition, our analysis was limited by the inability to assess vitamin D supplement use, as these supplements are generally not PBS-subsidized in Australia and therefore not available in the linked pharmaceutical data. Since vitamin D supplementation is a key determinant of vitamin D status, understanding supplementation trends could provide valuable insights into the appropriateness and drivers of vitamin D testing. Furthermore, information for some testing indications specified by the MBS was not available in our dataset (see Appendix A); therefore, we may have misclassified some tests as being without indication. However, these missing indications mostly occur rarely (e.g., gastrointestinal malabsorption) so the misclassification is likely to have been small.

## 5. Conclusions

In conclusion, this study provides compelling evidence of widespread and sustained growth in vitamin D testing in Australia. Coordinated initiatives are warranted to ensure vitamin D testing is more appropriately targeted to high-risk populations, improving resource allocation while maintaining optimal patient care. Since vitamin D supplementation is safe, effective, and relatively cheap, it may be a better use of healthcare resources to initiate supplementation, in the absence of testing, in people without clinical indications but who receive minimal sun exposure and/or have a high body mass index.

## Figures and Tables

**Figure 1 nutrients-17-02549-f001:**
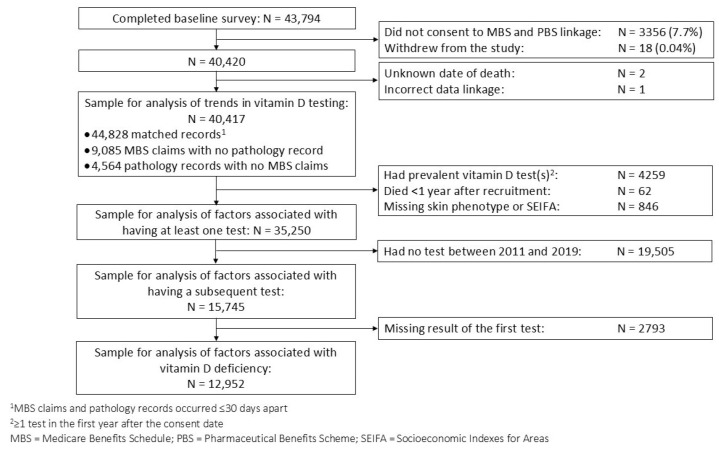
Flowchart of participant selection for main analyses.

**Figure 2 nutrients-17-02549-f002:**
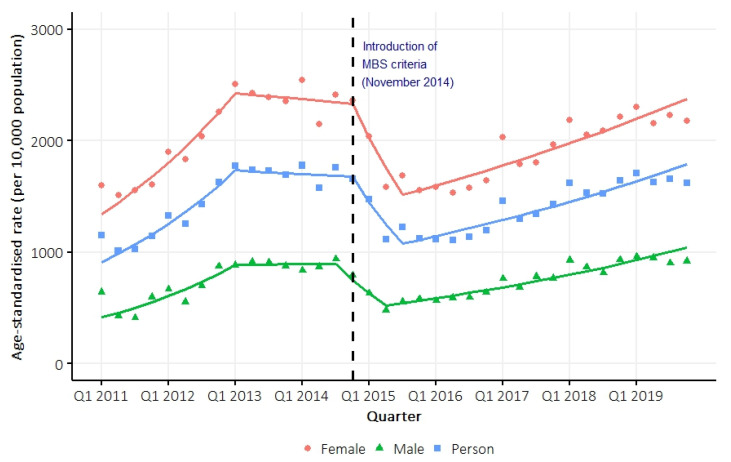
Overall and sex-specific age-standardized (Australian population 2001) person-based incidence rate of vitamin D testing between 2011 and 2019. Trend lines were estimated using Joinpoint regression models.

**Figure 3 nutrients-17-02549-f003:**
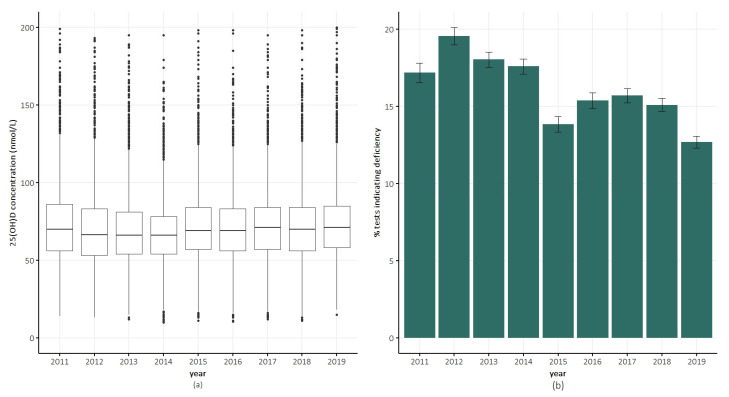
Trends in serum 25(OH)D concentrations and vitamin D deficiency in QSkin participants over time. (**a**) Box plots of serum 25(OH)D concentration. (**b**) Percentage of tests indicated vitamin D deficiency (serum 25(OH)D concentration <50 nmol/L) in a calendar year. Error bars indicate ±1 standard error.

**Table 1 nutrients-17-02549-t001:** Associations between participant characteristics and having at least one vitamin D test between 2011 and 2019.

Characteristic	*n* (%)	IR	HR (95% CI)	
			Crude Model	Adjusted Model ^1^
Sex				
Male	16,880 (47.9)	493	Ref	Ref
Female	18,370 (52.1)	1223	2.42 (2.34, 2.50)	2.52 (2.44, 2.61)
Age at cohort entry (years)				
<45	2859 (8.1)	660	Ref	Ref
45 to <50	4986 (14.1)	710	1.07 (1.00, 1.16)	1.10 (1.02, 1.18)
50 to <55	6375 (18.1)	760	1.15 (1.07, 1.23)	1.22 (1.13, 1.30)
55 to <60	6659 (18.9)	828	1.25 (1.16, 1.34)	1.35 (1.26, 1.45)
60 to <65	6516 (18.5)	873	1.31 (1.22, 1.41)	1.48 (1.38, 1.58)
65+	7855 (22.3)	1001	1.49 (1.40, 1.60)	1.75 (1.64, 1.87)
Ancestry origin				
White European	32,882 (94.1)	815	Ref	Ref
Other ^2^	2054 (5.9)	1008	1.23 (1.15, 1.31)	1.21 (1.13, 1.29)
Missing	314			
SEIFA category at baseline ^3^				
1—Most disadvantaged	7306 (20.7)	728	Ref	Ref
2	7300 (20.7)	775	1.06 (1.01, 1.12)	1.09 (1.04, 1.15)
3	7082 (20.1)	839	1.15 (1.09, 1.21)	1.19 (1.13, 1.25)
4	6807 (19.3)	873	1.19 (1.14, 1.26)	1.27 (1.20, 1.33)
5—Least disadvantaged	6755 (19.2)	942	1.29 (1.22, 1.35)	1.41 (1.34, 1.48)
BMI category at baseline				
Underweight	341 (1.0)	1281	1.36 (1.18, 1.56)	1.15 (1.00, 1.32)
Normal weight	11,254 (33.0)	929	Ref	Ref
Overweight	13,453 (39.5)	723	0.78 (0.76, 0.81)	0.93 (0.89, 0.96)
Obese	9041 (26.5)	845	0.91 (0.88, 0.95)	0.99 (0.95, 1.03)
Missing	1161			
History of regular smoking at baseline				
Never	19,093 (54.4)	866	Ref	Ref
Past	12,578 (35.8)	793	0.92 (0.89, 0.95)	1.01 (0.98, 1.05)
Current	3441 (9.8)	740	0.86 (0.81, 0.91)	0.97 (0.92, 1.03)
Missing	138			
Number of alcoholic drinks per week at baseline				
None	6529 (18.6)	996	Ref	Ref
<1	5765 (16.4)	949	0.95 (0.91, 1.00)	1.01 (0.96, 1.06)
2–4	6304 (18.0)	875	0.88 (0.84, 0.93)	0.98 (0.93, 1.03)
5–6	4592 (13.1)	807	0.82 (0.77, 0.86)	0.93 (0.88, 0.99)
7–13	5711 (16.3)	756	0.77 (0.73, 0.81)	0.91 (0.87, 0.96)
14+	6179 (17.6)	601	0.61 (0.58, 0.65)	0.88 (0.83, 0.93)
Missing	170			
Sun exposure in the year prior to baseline				
Low (≤3.5 h/week)	5004 (15.3)	1145	Ref	Ref
Medium (>3.5 to ≤10 h/week)	7000 (21.4)	895	0.79 (0.75, 0.83)	0.94 (0.89, 0.99)
High (>10 to ≤25 h/week)	14,556 (44.5)	782	0.69 (0.66, 0.73)	0.87 (0.83, 0.91)
Very high (>25 h/week)	6115 (18.7)	586	0.52 (0.49, 0.55)	0.83 (0.78, 0.88)
Missing	2575			
Sunscreen use in the year prior to baseline				
Never	7249 (20.7)	797	Ref	Ref
Less than 50% of the time	14,985 (42.7)	801	1.01 (0.96, 1.05)	0.97 (0.93, 1.01)
More than 50% of the time	9646 (27.5)	849	1.06 (1.02, 1.11)	0.98 (0.93, 1.02)
All the time	3183 (9.1)	955	1.19 (1.12, 1.27)	1.00 (0.94, 1.07)
Missing	187			
Number of GP visits in the 12 months before cohort entry				
0	3350 (9.5)	428	Ref	Ref
1	3442 (9.8)	604	1.40 (1.29, 1.53)	1.30 (1.19, 1.42)
2+	28,458 (80.7)	914	2.09 (1.96, 2.24)	1.61 (1.50, 1.73)
Rx-Risk comorbidity index at cohort entry (unweighted)				
0	18,718 (53.1)	689	Ref	Ref
1	6828 (19.4)	884	1.27 (1.22, 1.33)	1.24 (1.19, 1.30)
2+	9704 (27.5)	1096	1.56 (1.51, 1.62)	1.47 (1.41, 1.53)
Skin phenotype (predisposition to skin cancers) ^4^				
Lowest risk	12,461 (35.4)	789	Ref	Ref
Medium risk	11,238 (31.9)	806	1.02 (0.98, 1.06)	0.98 (0.94, 1.02)
Highest risk	11,551 (32.8)	809	1.12 (1.08, 1.17)	1.03 (0.99, 1.07)
Skin cancer excision prior to baseline (self-report)				
None	21,066 (60.2)	801	Ref	Ref
1	4879 (13.9)	888	1.10 (1.06, 1.16)	1.06 (1.01, 1.11)
2–10	7504 (21.4)	875	1.09 (1.05, 1.13)	1.10 (1.06, 1.15)
>10	1563 (4.5)	750	0.94 (0.86, 1.01)	1.03 (0.95, 1.12)
Missing	238			
Skin cancer cryotherapy prior to baseline (self-report)				
None	16,029 (45.7)	776	Ref	Ref
1–5	9164 (26.1)	894	1.15 (1.10, 1.19)	1.08 (1.04, 1.12)
6–10	3282 (9.4)	899	1.15 (1.09, 1.22)	1.12 (1.06, 1.19)
>10	6604 (18.8)	820	1.05 (1.01, 1.10)	1.09 (1.04, 1.14)
Missing	171			
Diagnosed with in situ melanoma subsequent to baseline				
No	34,332 (97.4)	829	Ref	Ref
Yes	918 (2.6)	679	0.91 (0.79, 1.04)	0.97 (0.85, 1.11)
Diagnosed with invasive melanoma subsequent to baseline				
No	34,739 (98.6)	827	Ref	Ref
Yes	511 (1.4)	844	1.12 (0.95, 1.32)	1.16 (0.99, 1.37)
Treated for keratinocyte cancer subsequent to baseline				
No	25,271 (71.7)	838	Ref	Ref
Yes	9979 (28.3)	785	1.04 (1.00, 1.08)	1.06 (1.02, 1.10)
Ever been prescribed with osteoporosis medication ^5^				
No	33,836 (96.0)	798	Ref	Ref
Yes	1414 (4.0)	2216	2.83 (2.64, 3.03)	1.96 (1.83, 2.11)
Recent osteoporosis medication ^5,6^				
No	34,415 (97.6)	822	Ref	Ref
Yes	835 (2.4)	5832	7.17 (5.90, 8.70)	5.32 (4.38, 6.46)
Ever been prescribed with antiepileptic medication ^5^				
No	34,283 (97.3)	821	Ref	Ref
Yes	967 (2.7)	1134	1.45 (1.31, 1.60)	1.41 (1.28, 1.56)
Recent antiepileptic medication ^5,6^				
No	34,644 (98.3)	826	Ref	Ref
Yes	606 (1.7)	2023	2.37 (1.62, 3.45)	2.35 (1.61, 3.43)
Ever been prescribed with menopausal hormone therapy ^5,7^				
No	15,045 (81.9)	1204	Ref	Ref
Yes	3325 (18.1)	1365	1.33 (1.25, 1.41)	1.25 (1.18, 1.32)
Recent menopausal hormone therapy ^5,6,7^				
No	15,555 (84.7)	1221	Ref	Ref
Yes	2815 (15.3)	1575	1.41 (1.16, 1.71)	1.36 (1.12, 1.65)

HR = hazard ratio, CI = confidence interval, IR = incidence rate (per 10,000 person-years), ref = reference category, SEIFA = Socio-Economic Indexes for Area. Baseline was at the time of recruitment to the QSkin Study. Cohort entry for this analysis was 12 months after recruitment. ^1^ All variables were adjusted for sex and age at cohort entry (with the exception of menopausal hormone therapy, which was not adjusted for sex). Additional adjustments were as follows: SEIFA was adjusted for skin phenotype; BMI was adjusted for SEIFA; number of GP visits was adjusted for skin phenotype, SEIFA, and Rx-Risk index at cohort entry; menopausal hormone therapy was adjusted for skin phenotype and SEIFA; other variables were adjusted for skin phenotype and SEIFA. ^2^ Includes the following: Aboriginal and Torres Islander; Maori; South Sea Islander; Asian; African/Caribbean; and mixed. ^3^ SEIFA score was calculated for all QSkin participants based on their postcode at baseline, then categorized into quintiles. ^4^ A skin phenotype score was calculated based on skin color, propensity to burn, propensity to tan, and natural hair color. We categorized the score into tertiles based upon the distribution of the score in QSkin participants. ^5^ Prior to the first test. ^6^ Events occurring within 90 days of first prescription considered. ^7^ Restricted to women.

**Table 2 nutrients-17-02549-t002:** Distribution of most likely indication for the first vitamin D test and result of the test.

Indication ^1^	% ^2^	Mean 25(OH)D (SD) ^3^	*n* (%) Deficient ^3^
No apparent indication	56.4	69.9 (22.2)	1105 (16.4)
Obesity (body mass index ≥30 kg/m^2^)	22.3	62.7 (20.9)	697 (26.2)
Low sun exposure (≤3.5 h/week outdoors)	18.6	61.8 (21.7)	656 (29.8)
Long-term glucocorticoid use ^4^	1.7	63.4 (21.2)	53 (26.5)
Recently prescribed with osteoporosis/antiepileptic medication ^5^	0.9	65.9 (23.2)	29 (25.9)

^1^ Each participant was assigned only one indication for the vitamin D test. If a participant had two or more indications, the highest-ranked indication was assigned to that participant. The hierarchy in which we assigned the indication was osteoporosis/antiepileptic medication, long-term glucocorticoid use, low sun exposure, and being obese. ^2^ Includes participants who were not missing any data required to determine whether the test was indicated (*n* = 14,463). ^3^ Includes participants who were not missing the result of the first vitamin D test, or any data required to determine whether the test was indicated (total *n* = 11,921, no apparent indication, *n* = 6746, at least one indication, *n* = 5175). ^4^ Medication with Anatomical Therapeutic Chemical code H02AB dispensed at least 4 times in the 12 months before the first vitamin D test. ^5^ Prescribed within 90 days before the test.

**Table 3 nutrients-17-02549-t003:** Associations between participant characteristics and vitamin D deficiency ^1^, based on the first test for each participant (*n* = 12,952 ^2^).

Characteristics	*n*	Median 25-Hydroxyvitamin D (25th, 75th Percentile)	% Deficient ^1^	Crude RR(95% CI)	Adjusted RR (95% CI) ^3^
Sex					
Male	4240	68 (55, 83)	17.0	Ref	Ref
Female	8712	64 (50, 77)	23.4	1.38 (1.28–1.49)	1.38 (1.27–1.49)
Age at cohort entry (years)					
<45	839	64 (50, 79)	23.1	1.05 (0.91–1.22)	1.02 (0.88–1.18)
45 to <50	1614	64 (50, 77)	24.0	1.09 (0.97–1.22)	1.08 (0.96–1.21)
50 to <55	2211	64 (51.5, 78.5)	22.0	Ref	Ref
55 to <60	2409	65 (52, 78)	21.6	0.98 (0.88–1.09)	0.99 (0.88–1.10)
60 to <65	2539	66 (52, 79)	21.3	0.97 (0.87–1.08)	0.99 (0.89–1.11)
65+	3340	66 (53, 81)	18.9	0.86 (0.77–0.95)	0.90 (0.81–1.00)
Ancestry origin					
White European	12,028	66 (52, 80)	20.3	Ref	Ref
Other ^4^	797	58 (44, 72)	35.9	1.77 (1.60–1.95)	1.66 (1.51–1.84)
SEIFA category at baseline ^5^					
1—Most disadvantaged	2430	67 (54, 80)	19.3	Ref	Ref
2	2580	65 (52, 79)	20.0	1.04 (0.93–1.16)	1.01 (0.91–1.13)
3	2616	66 (53, 81)	19.9	1.03 (0.92–1.16)	0.99 (0.89–1.11)
4	2577	65 (51, 79)	22.2	1.15 (1.03–1.29)	1.11 (0.99–1.23)
5—Least disadvantaged	2749	63 (50, 77)	24.8	1.29 (1.16–1.43)	1.25 (1.12–1.38)
BMI category at baseline					
Underweight	168	70 (55, 89)	17.3	1.05 (0.75–1.48)	1.03 (0.74–1.44)
Normal weight	4445	69 (55, 82)	16.4	Ref	Ref
Overweight	4536	66 (53, 79)	19.6	1.20 (1.10–1.31)	1.30 (1.19–1.43)
Obese	3374	60 (47, 74)	29.8	1.82 (1.67–1.98)	1.95 (1.80–2.12)
History of regular smoking at baseline					
Never	7326	65 (52, 79)	20.7	Ref	Ref
Past	4465	66 (52, 80)	20.4	0.98 (0.91–1.06)	1.03 (0.96–1.11)
Current	1113	62 (47, 76)	28.9	1.39 (1.26–1.54)	1.45 (1.31–1.60)
Number of alcoholic drinks per week at baseline					
None	2691	63 (48, 77)	26.5	Ref	Ref
<1	2321	62 (50, 76)	24.2	0.91 (0.83–1.00)	0.89 (0.81–0.98)
2–4	2417	66 (54, 80)	17.6	0.66 (0.60–0.74)	0.65 (0.59–0.73)
5–6	1649	67 (53, 81)	18.7	0.70 (0.63–0.79)	0.71 (0.63–0.80)
7–13	2010	68 (54, 82)	18.1	0.68 (0.61–0.76)	0.70 (0.63–0.78)
14+	1797	66 (53, 81)	20.8	0.78 (0.70–0.87)	0.86 (0.77–0.97)
Sun exposure in the year prior to baseline					
Low (≤3.5 h/week)	2260	60 (46, 75)	30.1	Ref	Ref
Medium (>3.5 to ≤10 h/week)	2742	64 (50, 76)	24.0	0.80 (0.73–0.87)	0.79 (0.72–0.87)
High (>10 to ≤25 h/week)	5180	67 (54, 80)	18.6	0.62 (0.57–0.67)	0.64 (0.59–0.70)
Very high (>25 h/week)	1737	69 (57, 84)	13.9	0.46 (0.40–0.53)	0.52 (0.45–0.60)
Sunscreen use in the year prior to baseline					
Never	2348	64 (50, 79)	24.4	Ref	Ref
Less than 50% of the time	5054	65 (52, 79)	21.5	0.88 (0.81–0.96)	0.82 (0.75–0.90)
More than 50% of the time	3388	66 (54, 80)	18.2	0.75 (0.68–0.83)	0.67 (0.60–0.74)
All the time	1206	64 (50, 80)	22.7	0.93 (0.82–1.06)	0.79 (0.70–0.90)
Number of GP visits in the 12 months before cohort entry					
0	711	65 (52, 80)	21.4	Ref	Ref
1	1007	65 (52, 78)	20.8	0.97 (0.81–1.17)	0.94 (0.78–1.13)
2+	11,234	65 (52, 79)	21.4	1.00 (0.86–1.16)	0.98 (0.85–1.14)
Rx-Risk comorbidity index (unweighted) at cohort entry					
0	6073	66 (52, 80)	20.4	Ref	Ref
1	2654	66 (52, 79)	20.6	1.01 (0.92–1.10)	1.07 (0.98–1.17)
2+	4225	64 (51, 78)	23.1	1.13 (1.05–1.22)	1.34 (1.23–1.46)
Skin phenotype (predisposition to skin cancers) ^6^					
Lowest risk	4407	66 (52, 80)	20.4	Ref	Ref
Medium risk	4061	66 (53, 80)	20.1	0.99 (0.91–1.07)	0.98 (0.90–1.07)
Highest risk	4484	64 (50, 77)	23.4	1.15 (1.06–1.24)	1.11 (1.03–1.20)
Skin cancer excision prior to baseline (self-report) ^7^					
No	7497	64 (51, 78)	22.7	Ref	Ref
Yes	5357	66 (53, 80)	19.4	0.86 (0.80–0.92)	0.88 (0.82–0.94)
Missing	98				
Skin cancer cryotherapy prior to baseline (self-report) ^7^					
No	5495	64 (50, 77)	23.9	Ref	Ref
Yes	7381	66 (53, 80)	19.4	0.81 (0.76–0.87)	0.81 (0.76–0.87)
Missing	76				
Diagnosed with in situ melanoma subsequent to baseline ^7^					
No	12,763	65 (52, 79)	21.4	Ref	Ref
Yes	189	69 (56, 81)	16.4	0.77 (0.55–1.06)	0.83 (0.61–1.14)
Diagnosed with invasive melanoma subsequent to baseline ^7^					
No	12,828	65 (52, 79)	21.3	Ref	Ref
Yes	124	69 (54, 82)	18.5	0.87 (0.60–1.26)	0.96 (0.66–1.38)
Treated for keratinocyte cancer subsequent to baseline ^7^					
No	10,205	65 (51, 79)	22.1	Ref	Ref
Yes	2747	67 (54, 81)	18.3	0.83 (0.76–0.90)	0.88 (0.81–0.97)
Osteoporosis medication ^7^					
No	12,251	65 (52, 79)	21.6	Ref	Ref
Yes	701	69 (56, 85)	15.5	0.72 (0.60–0.86)	0.72 (0.61–0.87)
Antiepileptic medication ^7^					
No	12,621	65 (52, 79)	21.1	Ref	Ref
Yes	331	61 (46, 76)	29.6	1.40 (1.18–1.66)	1.49 (1.26–1.76)
Menopausal hormone therapy ^7,8^					
No	7484	63 (50, 77)	24.1	Ref	Ref
Yes	1228	66 (53, 78)	19.5	0.81 (0.72–0.91)	0.82 (0.73–0.93)

25(OH)D = 25-hydroxyvitamin D, RR = risk ratio, CI = confidence interval, ref = reference category, SEIFA = Socioeconomic Index for Areas, BMI = body mass index. ^1^ Serum 25(OH)D <50 nmol/L. ^2^ Sample restricted to those with 25(OH)D results available. In some models, the sample was smaller due to missingness of the explanatory variable. ^3^ Adjusted RRs were estimated using log-binomial models. All variables were adjusted for sex, age at cohort entry, and testing laboratory (with the exception of menopausal hormone therapy which was not adjusted for sex). Additional adjustments were as follows: SEIFA was adjusted for skin phenotype; BMI was adjusted for SEIFA; sunscreen use was adjusted for skin phenotype, SEIFA, and sun exposure in the year prior to baseline; menopausal hormone therapy was adjusted for skin phenotype and SEIFA; and other variables were adjusted for skin phenotype and SEIFA. ^4^ Participants whose ancestry identified as Aboriginal and Torres Islander, Maori, South Sea Islanders, Asian, from African or Caribbean countries, or mixed between those groups. ^5^ SEIFA (Socio-Economic Indexes for Area) score was produced for all QSkin participants based on their postcode at baseline, then categorized into quintiles. ^6^ A skin phenotype score was calculated based on skin color, propensity to burn, propensity to tan, and natural hair color. We categorized the score into tertiles based upon the distribution of the score in QSkin participants. ^7^ Ever having been exposed to the medication/treatment or diagnosed before the test. ^8^ Restricted to women.

**Table 4 nutrients-17-02549-t004:** Associations between participant characteristics and repeat testing among people who had at least one test between 2011 and 2019 (*n* = 15,745).

Characteristic	HR (95% CI)	
	Crude Model	Adjusted Model ^1^
Sex		
Male	Ref	Ref
Female	1.27 (1.22, 1.33)	1.25 (1.20, 1.29)
Age at time of the preceding test (years)		
<45	Ref	Ref
45 to <50	1.23 (1.07, 1.42)	1.18 (1.03, 1.36)
50 to <55	1.28 (1.11, 1.47)	1.21 (1.05, 1.38)
55 to <60	1.38 (1.20, 1.59)	1.28 (1.12, 1.47)
60 to <65	1.46 (1.26, 1.68)	1.32 (1.16, 1.51)
65+	1.60 (1.39, 1.84)	1.45 (1.26, 1.65)
GP visits per year before the preceding test		
0–2	Ref	Ref
3–4	1.10 (1.05, 1.16)	1.04 (0.99, 1.09)
5–6	1.17 (1.11, 1.23)	1.08 (1.03, 1.14)
>6	1.45 (1.39, 1.52)	1.24 (1.18, 1.30)
Rx-Risk index (unweighted) at time of the preceding test		
0	Ref	Ref
1	1.10 (1.06, 1.15)	1.06 (1.02, 1.11)
2+	1.28 (1.24, 1.33)	1.17 (1.12, 1.21)
Result of previous vitamin D test (nmol/L) ^2^		
<50	1.38 (1.33, 1.44)	1.46 (1.40, 1.53)
50 to <75	Ref	Ref
75+	0.97 (0.94, 1.01)	0.93 (0.90, 0.96)

CI = confidence interval, GP = general practitioner, HR = hazard ratio, ref = reference category. ^1^ All models included age, sex, and the number of previous vitamin D tests. The model for number of GP visits also included Rx-Risk index at time of the preceding test, and the model for result of previous test also included the season of the preceding test. ^2^ Sample restricted to those for whom the test result was available (*n* = 12,950).

## Data Availability

The data underlying this article may be shared on request to the corresponding author, subject to ethical and regulatory clearances from the host institution.

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
