# Peer review of "Factors Associated with Vitamin D Testing: A Population-Based Cohort Study in Queensland, Australia"

_nutrients, 2025, doi:10.3390/nu17152549_

Round 1

Reviewer 1 Report

Comments and Suggestions for Authors

In this paper authors examine trends and factors associated with vitamin D testing in Australia using a large cohort study. Overall, I found the paper well-written, methodological approach strong and exhaustive, and presented results relevant and abundant. I have a few comments that might strengthen the paper further.

Major comments

  1. Vitamin D supplementation is not really mentioned; however this is a major determinant of vitamin D status. Given linkage with PBS, is it possible to get data on vitD supplementation? It would be interesting to see whether supplement taking is linked with more/less testing. At a minimum, the absence of this factor in the study should be acknowledged and discussed.
  2. I was also surprised to note that month or season of testing have not really been considered. Could authors at least provide basic results for this? Particularly when looking at the prevalence of deficient results amongst all tested.

Minor comments

  1. Section “2.2 Outcomes” rewrite for clarity. Eg “incidence of at least one test…” etc.
  2. Consider giving an explanation to explain what might underpin the fact that 17% of testing episodes were not matched with a lab result (line 94). How does this map onto Figure 1 (is this N=9058?) What was done when pathology record was found, but MBS claim was not?
  3. Briefly explain what Rx-risk comorbidity index is (line 101)
  4. Indication – for completeness, it is really important to list indications for which data were not available also. This will help the reader interpret findings that most tests were done without indication. Check throughout, e.g. Line 205– If I understand this correctly, information about all indications for testing was not available for the study population? When discussing absence of indication it is a bit misleading if this is ignored (ie. If there was an indication authors had no data on) – rephrase please. In discussion, make sure to make reference to this. In discussion, need to clearly state that adding Rx-Risk as an indication changed results substantially
  5. Results – give exact N of participants who had: 0, 1, 2, 3, 4, 5, 6-10, =>11 vitD tests. Anything to note for people who had >5 tests?
  6. Table 1 – consider showing statistically significant results in bold
  7. Figure 2 – introduction of the new eligibility criteria in Nov 2014 is such a major event – consider adding to the figure. For Discussion: Is there anything that might have caused the late 2012 change in direction, or the bounce-back mid-2015?
  8. In supplementary material, include histogram of 25OHD by month of sample.
  9. Report number of high 25OHD test results (e.g. >150 and >250 nmol/l)
  10. Table 2. Add Rx-Risk. Could mean 25OHD and % deficient be given separately for each indication also?
  11. Report correlation between first and second test where available (scatter plot in supplementary would be good to)
  12. Discussion around test done privately would be helpful, ie those not government-reimbursed. How likely is this in Australia?
  13. Is it known what assay was used to measure 25OHD? (I expect it will be a mix of methods.) Brief discussion of this would be useful and consideration how this might affect findings
  14. It would be insightful to introduce the idea that in some cases GP may be checking that 25OHD is not too high
  15. In the introduction, it is suggested that no studies investigated testing among individuals who avoid sun – it would be good to pick up this point in the discussion
  16. Consider discussing your finding that proportion of deficient test results decreased
  17. Line 309 – is it appropriate to say “only” for 30% prevalence of deficiency among those who avoid sun?

Reviewer 2 Report

Comments and Suggestions for Authors

Strengths of the study:

  • The study is based on a large, well-established cohort (QSkin), which enhances the generalizability of the findings.
  • The linkage with administrative data (MBS, PBS, pathology) is a significant strength, allowing for robust analyses of testing behavior and outcomes.
  • The study highlights real-world patterns of potentially inappropriate vitamin D testing, an issue relevant to health policy and clinical practice both in Australia and internationally.
  • Statistical methods used (e.g., Cox regression, Joinpoint regression) are appropriate and well-described.

Weaknesses and areas for improvement:

  • For methodology: The classification of “indication” for testing is largely based on available administrative data. However, it remains unclear how comprehensive this proxy strategy is. Could the authors discuss more explicitly which known clinical indications could not be assessed and how that might have impacted the estimates of "no indication" testing? Furthermore, you should consider providing more clarity on the sensitivity analysis that includes the Rx-Risk score. The rationale for inserting it into the hierarchy and its potential limitations should be discussed.
  • For interpretation of data: The conclusion that many tests were conducted without indication is compelling, but some caution is needed given possible unmeasured clinical justifications. A discussion on how this limitation affects the strength of conclusions would enhance the paper. On the other hand, the interpretation that repeat testing in those without deficiency is inappropriate should be more nuanced. Could some of these reflect seasonal monitoring or changes in patient risk status?

Comments and suggestions for authors

I recommend to review these ideas:

  1. The abbreviation list is helpful, but “SEIFA” should be fully defined in the abstract for international readers.
  2. Minor editorial suggestions:
  • Line 28: "more appropriately" → consider rephrasing to "in alignment with clinical guidelines".
  • Line 60: "describe trends" → "analyze trends" may be more precise.
